# NOCTA: Non-Greedy Objective Cost-Tradeoff Acquisition for Longitudinal Data

## Abstract

In many critical applications, resource constraints prevent observing all features at test time, motivating selective information acquisition for the predictions. For example, in healthcare, patient data spans diverse features ranging from lab tests to imaging studies, each may carry different information and must be acquired at a cost of time, money, or risk to the patient. Moreover, temporal prediction tasks, where both instance features and labels evolve over time, introduce additional complexity in deciding when or what information is important. In this work, we propose NOCTA, a **N**on-Greedy **O**bjective **C**ost-**T**radeoff **A**cquisition method that sequentially acquires the most informative features at inference time while accounting for both temporal dynamics and acquisition cost. We first introduce a cohesive estimation target for our NOCTA setting, and then develop *two complementary* estimators: 1) a *non-parametric method* based on nearest neighbors to guide acquisitions (NOCTA-NP), and 2) a *parametric method* that directly predicts the utility of potential acquisitions (NOCTA-P). Experiments on synthetic and real-world medical datasets demonstrate that both NOCTA variants outperform existing baselines, achieving higher accuracy at lower acquisition costs.

## 1 Introduction

In many real-world scenarios, acquiring features at inference time comes at a cost. For example, in clinical settings (a driving application for this work), diagnostic decisions often rely on sequentially gathering information such as lab tests or imaging. Each type of data acquisition comes at the cost of time, financial expense, or risk to the patient. In contrast, many of the existing machine learning frameworks assume that *all* features are freely available at the outset. In this work, we focus on a longitudinal Active Feature Acquisition (AFA) (Saar-Tsechansky et al., 2009; Kossen et al., 2022) setting, where a model is responsible for (1) sequentially deciding *which features* to acquire and at *what time* during inference, balancing their utility against acquisition costs, and (2) *making predictions* with the partially observed features it has chosen to acquire.

In particular, we focus on a *longitudinal* setting, where long temporal contexts may be crucial for both selecting valuable feature acquisitions and making accurate predictions. Fig. 1 illustrates an example of longitudinal AFA methods driven by an autonomous agent operating in a clinical scenario. At each visit, the agent (1) reviews previously collected data alongside any measurement acquired during the current visit, (2) predicts the patient's current status, and (3) recommends a future follow-up plan by specifying both the visit interval from a predefined discrete time point (e.g., 3 or 6 months) and the informative tests to perform at the visit, while accounting for acquisition costs. Although in principle a comprehensive set of tests could be performed, practical constraints such as time, cost, and resource limitations necessitate a more selective approach. As a result, the agent prioritizes the most informative acquisitions. Between the visits, the agent projects the patient's health trajectory to future time points, using the data acquired so far and the length of the proposed time interval. Because the agent cannot revisit earlier time points, it must carefully decide whether to defer or perform tests at each time step to avoid missing potentially important acquisitions.

This longitudinal AFA setting presents challenges, including: deciding which features to acquire at each visit, facilitating early predictions for timely interventions, and accounting for temporal settings where missed acquisitions at earlier time points are permanently inaccessible. Although previous work has explored various approaches for longitudinal AFA, some methods (Kossen et al., 2022)

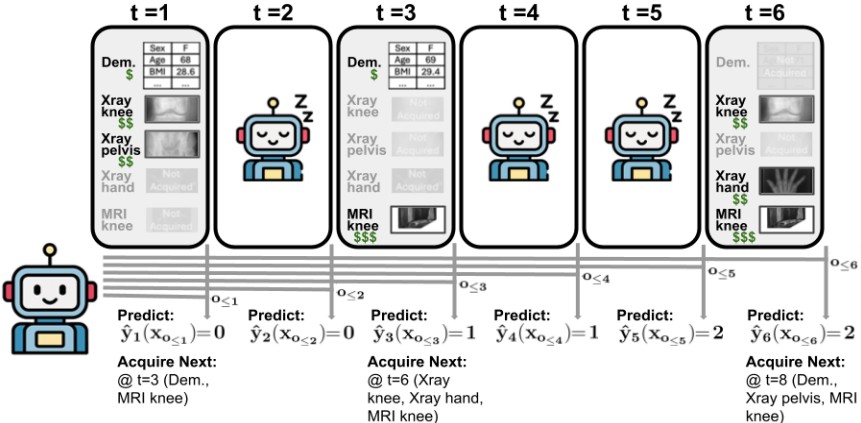

Figure 1: Illustration of longitudinal active feature acquisition in a clinical scenario: at each time point, an autonomous agent reviews previously collected and newly acquired data, predicts the patient's current status, and recommends a subset of examinations for the next visit. This process is repeated until no further follow-up is recommended by the agent.

make a single prediction across all time points, neglecting dynamic predictions, which are critical in clinical settings to detect deterioration early and adjust treatment plans (Qin et al., 2024). Furthermore, recent reinforcement learning (RL)-based methods (Qin et al., 2024; Nguyen et al., 2024) often face optimization challenges, while greedy strategies (Ghosh and Lan, 2023; Gadgil et al., 2024) risk suboptimal decisions by failing to anticipate the joint information of future acquisitions.

**Contributions.** In this work, we propose NOCTA, **N**on-Greedy **O**bjective **C**ost-**T**radeoff **A**cquisition, a non-greedy method that explicitly addresses longitudinal AFA challenges without training an RL policy, while maintaining an effective acquisition strategy. Firstly, we introduce the estimation target of NOCTA that balances prediction accuracy against the cost of acquiring features. Secondly, we develop two alternative approaches within the NOCTA framework: (1) a non-parametric approach that leverages nearest neighbors to guide the acquisition (NOCTA-NP), and (2) a parametric approach that directly estimates the utility of future acquisitions (NOCTA-P). Thirdly, we evaluate our method on both synthetic and real-world medical datasets, demonstrating NOCTA consistently outperforms state-of-the-art baselines while achieving a lower acquisition cost.

## 2 RELATED WORK

**Active Feature Acquisition.** Prior work in AFA (Saar-Tsechansky et al., 2009; Sheng and Ling, 2006) studies the trade-off between prediction and feature acquisition cost in a non-longitudinal context. Recent works (Shim et al., 2018; Yin et al., 2020; Janisch et al., 2020) frame AFA as an RL problem; however, they are challenging to train in practice due to the complicated state and action space with dynamically evolving dimensions. Other works (Li and Oliva, 2021) use surrogate generative models, which model multidimensional distributions, or employ greedy policies that ignore joint informativeness (Covert et al., 2023; Ma et al., 2018; Gong et al., 2019). To overcome these issues, Valancius et al. (2023) employs a non-parametric oracle-based method, and Ghosh and Lan (2023) learn a differentiable acquisition policy by jointly optimizing selection and prediction models through end-to-end training. However, these works address AFA in a non-temporal context, whereas many real-world tasks require sequential decisions over time, motivating the longitudinal AFA.

**Longitudinal Active Feature Acquisition.** Longitudinal AFA extends the standard AFA to include the temporal dimension. In this setting, a policy must account for temporal constraints where past time points can no longer be accessed. As a result, a policy must decide which features to acquire and determine the optimal timing for each acquisition. For instance, Zois and Mitra (2017) derives dynamic-programming and low-complexity myopic policies for finite-state Markov chains, ASAC (Yoon et al., 2019) uses actor-critic to jointly train feature selection and predictor network for prediction. Additionally, Nguyen et al. (2024) uses RL to optimize acquisition timing but acquires all available features at the selected time point. Recently, Qin et al. (2024) prioritizes timely detec-

tion by allowing flexible follow-up intervals in continuous-time settings, ensuring that predictions are timely and accurate. However, RL-based approaches can be challenging to train due to issues such as sparse rewards, high-dimensional observation spaces, and complex temporal dependencies (Li and Oliva, 2021). Rather than relying on RL to learn a policy, our proposed approach, NOCTA, explicitly models and optimizes the trade-off between predictive accuracy and acquisition cost.

# 3 METHODOLOGY

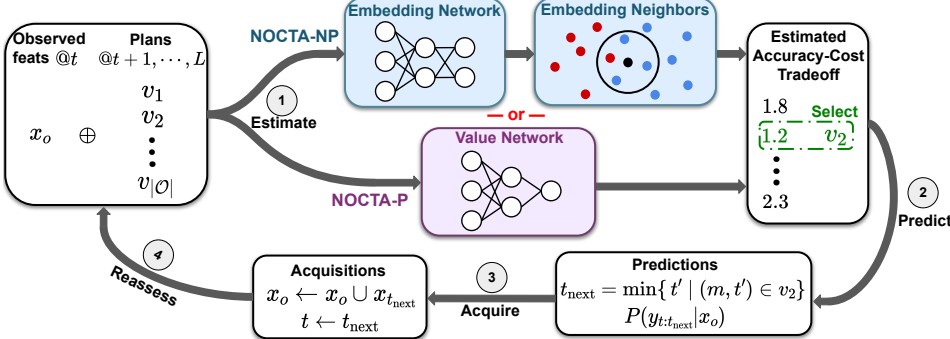

Figure 2: An overview of NOCTA. The framework starts at time $t$ with observed features $x_o$ and a set of $|\mathcal{O}|$ binary candidate masks $v$ of size $M \times (L - t)$, indicating plans of future features to acquire from time $t + 1$ to $L$. NOCTA provides two complementary estimators (NOCTA-NP and NOCTA-P) to predict the accuracy-cost tradeoff for each plan. After selecting the optimal plan, NOCTA predicts for the time points up to the next scheduled visit $t_{\text{next}}$, and acquires the corresponding features $x_{t_{\text{next}}}$, which are appended to $x_o$. After acquiring this new data, the entire process repeats and moves to the immediate next time step $t_{\text{next}} > t$. This allows NOCTA to dynamically reassess and revise its acquisition plan based on the most recent information rather than being locked into a static plan.

**Notation.** We formally define the longitudinal AFA setup: let $\mathcal{D} = \{(x_i, y_i)\}_{i=1}^N$ denote the dataset of $N$ instances (e.g., patients). For each $i$, $x_i$ is a temporal sequence with $L$ time steps, i.e., $x_i = \{x_{i,t}\}_{t=1}^L$, with $t \in \{1, \ldots, L\}$ indexing the time step. At each time step, $x_{i,t} \in \mathbb{R}$ includes measurements of $M$ (costly) features (note that it is possible to extend to individual acquisitions that yield multi-dimensional feature vectors for multi-modal cases), i.e., $x_{i,t} = \{x_{i,t}^m\}_{m=1}^M$. Note that it is possible for $x_{i,t}^m = \emptyset$, which indicates the feature is unobserved. In this work, we focus on a classification task where labels $y_{i,t} \in \{1, 2, \ldots, C\}$ are defined for each timepoint $t$, and $C$ represents the number of classes. Following Qin et al. (2024); Yoon et al. (2019), each feature $m$ has a time-invariant [1] cost $c^m$. Lastly, we use colons to represent selecting subsequences along a specific axis; for example, $x_{1:t} = (x_1, \ldots, x_t)$ represents the measurements from timepoint 1 to $t$.

## 3.1 LONGITUDINAL AFA MODELED AS AN MDP

Longitudinal AFA can be formulated as a Markov Decision Process (MDP). The state $s_t = (x_o, o, t)$, where $x_o$ denotes the (partial) observations before time $t$ and $o \subseteq \{(m, t') \mid m \in \{1, \ldots, M\}, t' \in \{1, \ldots, t-1\}\}$ records which features were acquired. We suppose access to a pre-trained arbitrary conditioning classifier $\hat{y}$ (Shim et al., 2018; Li and Oliva, 2021), that can predict the label at any timepoint $k$ with $x_o$: $\hat{y}_k(x_o)$. The action space is $a \in \{a_{t'}\}_{t'=t}^L \cup \{\varnothing\}$, where $a_{t'} \in \{0, 1\}^M$ indicates which of the $M$ features to acquire at the current or future time $t'$, or, when $a = \varnothing$, to terminate and predict for all remaining timepoints. For a non-terminal action, the transition is $(x_o, o, t) \rightarrow (x_{o \cup a_{t'}}, o \cup a_{t'}, t' + 1)$, where (for notational simplicity) $o \cup a_{t'}$ denotes the union of the previous acquisitions and tuples of features at time $t'$ in $a_{t'}$. Since this implies that we are not acquiring any information until time $t'$, the reward is the negative cross-entropy (CE) loss of making predictions up to $t'$ (i.e., $\hat{y}_t, \ldots, \hat{y}_{t'-1}$) with previous information ($x_o$) and respective negative CE loss and cost of acquiring new features at time

---

[1] NOCTA easily extends to time-varying costs by replacing $c^m$ with $c^{m,t}$ (cost of modality $m$ at time $t$), while leaving all other components unchanged.

$t'$: $-\sum_{k=t}^{t'-1} \text{CE}(\hat{y}_k(x_o), y_k) - \text{CE}(\hat{y}_{t'}(x_{o \cup a_{t'}}), y_{t'}) - \alpha \sum_{m=1}^{M} a_{t',m} c^m$, where $\alpha$ is an accuracy-cost tradeoff hyperparameter. When terminated ($a = \varnothing$) at timepoint $t$, we are not acquiring any additional information for the remaining timepoints, thus the reward is $-\sum_{k=t}^{L} \text{CE}(\hat{y}_k(x_o), y_k)$, which represents the negative cross-entropy loss accumulated from $t$ to $L$. We denote our policy as $\pi(x_o, o, t)$, which selects actions to maximize the expected cumulative rewards through $L$.

**Why is Longitudinal AFA MDP a Challenging Problem?** The challenges of longitudinal AFA with MDP include: 1) *large and complex action space*, which requires decisions on when to acquire and which features to acquire; 2) *complicated state space*, as it evolves dynamically as new features are captured over time; and 3) *credit assignment problem*, since acquisitions are awarded at an aggregate level, it is hard to disentangle exactly which of the acquisitions were most responsible for prediction rewards (Li and Oliva, 2021). Despite these, existing works (Qin et al., 2024; Yoon et al., 2019; Kossen et al., 2022; Nguyen et al., 2024) often rely on RL. In this work, we propose NOCTA, an RL-free approach that not only provides a theoretical lower bound for the optimal longitudinal AFA MDP (see Appx. A.5), but also empirically outperforms RL-based baselines.

## 3.2 NOCTA: GENERALIZED COST OBJECTIVE AND ACQUISITION ALGORITHM

We now introduce NOCTA through a novel objective that shall act as a proxy for the MDP value of states. Let $\mathcal{V} = \{(m, t') \mid m \in \{1, \ldots, M\}, t' \in \{t+1, \ldots, L\}\}$ represent the set of candidate feature-time pairs that are still available to acquire after time $t$. We ask: (1) *which subset in $\mathcal{V}$ should we acquire* to balance the tradeoff between improved accuracy and acquisition cost, or (2) *should we terminate and make the prediction* for the remaining time points based on what we have observed in $x_o$? That is, NOCTA determines the set of potential future acquisitions $u$:

$$u(x_o, o) = \underset{v \subseteq \mathcal{V}}{\arg\min} \, \mathbb{E}_{y_{t+1:L}, x_v | x_o} [\ell(x_o \cup x_v, y, t)] + \alpha \sum_{(m,t') \in v} c^m, \qquad (1)$$

where

$$\ell(x_o \cup x_v, y, t) = \sum_{k=t+1}^{L} \text{CE}(\hat{y}_k(x_o \cup x_{v_{t+1:k}}), y_k), \qquad (2)$$

represents the accumulated loss when only acquiring the set of future feature-time pairs $v$, and $x_{v_{t+1:k}}$ denotes the features in the candidate subset $v$ from time step $t+1$ to time step $k$. Because the full trajectories $(x_v, y_{t+1:L})$ are available for the training set $\mathcal{D}$, the expectation in Eq. (1) can be evaluated during training. Note that predictions can only rely on the current and previously captured data, but not on that which will be obtained in the future. The choice of $\alpha$ is application-specific: some applications allow a higher cost (i.e., low $\alpha$) and some are more cost-restricted (i.e., high $\alpha$). Intuitively, Eq. (1) non-greedily looks ahead at how acquiring a new set of feature-time pairs in $\mathcal{V}$ might reduce the classification loss from $t + 1$ to $L$, while penalizing the acquisition cost. For simplicity, we refer to feature-time pairs as "features" in the remainder of the paper.

When Eq. (1) returns $u(x_o, o) = \varnothing$, it indicates that the loss improvement does not justify the cost of acquiring a new feature, so we stop acquiring and predict for all subsequent time points using observed features $x_o$. Otherwise, if new features $u(x_o, o) = v \subseteq \mathcal{V}$ are acquired, the model predicts for the subsequent time points until the next scheduled acquisitions. This non-greedy policy for NOCTA directly balances the tradeoff between acquisition cost and predictive performance.

**Acquisition Algorithm.** The acquisition process of NOCTA is detailed in Algorithm 1. Intuitively, NOCTA mirrors the clinical workflow: order tests, read results, and reassess. When NOCTA selects the acquisition plan $\hat{u}(x_o, o)$, we do not commit to the entire sequence. Instead, we only execute the acquisition scheduled at the immediate next time point $t_{\text{next}}$, update the observed features $(x_o, o)$, and replan the acquisition schedule. Note that the model can also continue on with its original plan if it still believes those remaining acquisitions are optimal in light of the new acquired information in $t_{\text{next}}$. This iterative design reflects the principle that as new observed features alter our belief about the labels, it should also update our estimate of the utility of yet-to-be-acquired information.

**Theorem 1.** *(Informal) NOCTA lower bounds the value of the optimal longitudinal AFA MDP policy.*

In other words, acquiring according to the NOCTA objective (Eq.(1)) serves as an approximation to the optimum longitudinal AFA MDP (proof in Appx. A.5). However, evaluating this objective is intractable as in practice we do not have access to the true expectation of labels and unacquired

---

**Algorithm 1** NOCTA: Non-Greedy Objective Cost-Tradeoff Acquisition

---

**Require:** Observed features $o$ (possibly $o = \varnothing$), instance values $x_o$, number of features $M$, number of time steps $L$, cost for feature $m$ as $c^m$, tradeoff parameter $\alpha > 0$, estimator $\hat{y}_k$.
1: Initialize: predictions $\leftarrow []$; terminate $\leftarrow$ false; $t \leftarrow 0$
2: **while** not terminate **do**
3:      $\mathcal{V} \leftarrow \{(m, t') \mid m \in \{1, \dots, M\}, t' \in \{t+1, \dots, L\}\} \setminus o$
4:      $\hat{u}(x_o, o) \approx \underset{v \subseteq \mathcal{V}}{\arg\min} \Big[ \mathbb{E}_{y_{t+1:L}, x_v \mid x_o} \left[ \ell(x_o \cup x_v, y, t) \right] + \alpha \sum_{m \in v} c^m \Big]$
5:      **if** $\hat{u}(x_o, o) = \varnothing$ **then**
6:          $t_{\text{next}} \leftarrow L$; terminate $\leftarrow$ true
7:      **else**
8:          $t_{\text{next}} \leftarrow \min\{t' \mid (m, t') \in \hat{u}(x_o, o)\}$
9:          $o \leftarrow o \cup \{(m, t_{\text{next}}) \mid (m, t_{\text{next}}) \in \hat{u}(x_o, o)\}$
10:     **end if**
11:     **for** $t' = t+1, \dots, t_{\text{next}}$ **do**
12:         append $\hat{y}_{t'}\left(x_{o_{1:t'}}\right)$ to predictions
13:     **end for**
14:     $t \leftarrow t_{\text{next}}$
15: **end while**
16: **return** predictions

---

values for the test samples, $\mathbb{E}_{y_{t+1:L}, x_v \mid x_o}$ in Eq. (1). We therefore develop two ways to estimate $u(x_o, o)$: a non-parametric method that uses nearest neighbors on learned embedding space (Sec. 3.3) and a parametric method that directly predicts the accuracy-cost tradeoff (Sec. 3.4).

### 3.3 NOCTA-NP: NON-PARAMETRIC METHOD

In this subsection, we show how to estimate $u(x_o, o)$ using a non-parametric method, coined NOCTA-NP. The method uses the observed features and estimates Eq. (1) by finding similar cases in the training set identified via nearest neighbors. In order to assuage issues with the curse of dimensionality, we propose to compute nearest-neighbor distances using representations from an embedding network trained to encode information for future candidate selection.

**Estimating $u(x_o, o)$.** We first demonstrate the acquisition process through neighbors in the raw feature space. To estimate the optimal acquisition plan $u(x_o, o)$, we search for the candidate subset $v \in \mathcal{V}$ that minimizes the accuracy-cost tradeoff. For any given candidate plan $v$ at the time $t$, we estimate its accuracy-cost tradeoff by sampling labels and unacquired features through neighbors:

$$\mathbb{E}_{y_{t:L}, x_v \mid x_o} \left[ \ell(x_o \cup x_v, y, t) \right] \approx \frac{1}{K} \sum_{i \in N_K(x_o)} \sum_{k=t+1}^{L} \text{CE}(\hat{y}_k(x_{i,o} \cup x_{i,v_{1:k}}), y_{i,k}), \qquad (3)$$

where $N_K(x_o)$ represents the set of $K$-nearest neighbors ($K$-NN) in the dataset $\mathcal{D} = \{(x_i, y_i)\}_{i=1}^{N}$, with neighbors identified based on the observed features $o$ at the current time $t$. Specifically, the neighbors could be determined by comparing the distance $d(x_o, x_{i,o}) \to \mathbb{R}_+$, which directly uses the raw feature values, for example $d(x_o, x_{i,o}) = \|x_o - x_{i,o}\|_2$. *That is, Eq. (3) leverages the neighbors to approximate the expected cross-entropy loss for any candidate acquisition plan $v$.*

**Limitations of Raw Features.** Since $x_o$ accrues features through multiple time points, the raw features become high-dimensional, which impairs distance metrics due to the curse of dimensionality. To mitigate this, we propose computing distance within an *efficient embedding space* $\mathbb{R}^E$ learned by a network $g_\phi$, which is tailored to capture future candidate selection for longitudinal AFA task. This controlled, lower-dimensional space allows for a more effective nearest neighbor search.

**Embedding Network.** We first introduce the embedding network that maps partially observed features into a smaller embedding space where distances capture the similarity of their acquisition characteristics. Formally, we learn an embedding network $g_\phi : x_{i,o} \mapsto \mathbb{R}^E$ such that two instances $x_{i,o}$ and $x_{j,o}$ are mapped to nearby points in $\mathbb{R}^E$ if the additional (future) acquisitions available to them are expected to influence their accuracy-cost tradeoff in a similar manner. Conversely, instances with divergent accuracy-cost tradeoff are pushed away in $\mathbb{R}^E$.

To achieve this, we define a similarity score between instances based on their future-candidate distribution (detailed later in this section). Given two partially observed samples $x_{i,o}$ and $x_{j,o}$, we can quantify similarity between their respective future-candidate distribution, denoted as $\zeta_{i,o}$ and $\zeta_{j,o}$, using the exponential of their negative Jensen-Shannon (JS) divergence:

$$\mathtt{sim}_{ij} = \exp\big(-\beta \, \mathrm{JS}(\zeta_{i,o}, \zeta_{j,o})\big), \tag{4}$$

where $\beta > 0$ controls how strongly divergence affects similarity. Letting $\delta_{ij} = \big\|g_\phi(x_{i,o}) - g_\phi(x_{j,o})\big\|^2$ be the embedding distance, we minimize a contrastive-type loss (Chopra et al., 2005):

$$\mathcal{L}_{\mathrm{emb}} = \frac{1}{2}\Big(\mathtt{sim}_{ij} \cdot \delta_{ij} + (1 - \mathtt{sim}_{ij}) \cdot \max\big(0, \, \gamma - \delta_{ij}\big)^2\Big), \tag{5}$$

where $\gamma > 0$ is the margin parameter. Minimizing $\mathcal{L}_{\mathrm{emb}}$ encourages $g_\phi$ to create an embedding space where distance reflects the similarity of future-candidate distributions. Instead of applying $K$-NN in the raw high-dimensional feature space, we compute distance in $\mathbb{R}^E$ with controlled dimension $E \ll ML$. This reduction not only improves the effectiveness of distance metrics (we later show in the ablations) but also makes neighbor search more computationally efficient. That is, we find the $K$ neighbors of any observation $x_o$ using the embedding space distance $d(x_o, x_{i,o}) = \|g_\phi(x_o) - g_\phi(x_{i,o})\|_2$. Using these $K$ neighbors, we estimate Eq. (3) to derive the acquisition plan $u(x_o, o)$.

**Future-Candidate Distribution** $\zeta_o$**.** We now formally introduce the future-candidate distribution $\zeta_o$. For any partially observed features $x_o$ at time $t$ and the corresponding labels $y$ of the training dataset $\mathcal{D}$, we identify the top-$\kappa$ candidate subsets $v_1, \ldots, v_\kappa \subseteq \mathcal{V} = \{(m, t') \mid m \in \{1, \ldots, M\}, \, t' \in \{t+1, \ldots, L\}\}$, in ascending order of:

$$r_l = \ell(x_o \cup x_{v_l}, y, t) + \alpha \sum_{(m, t') \in v_l} c^m. \tag{6}$$

Eq. (6) quantifies the accuracy-cost tradeoff for candidate $v_l$, and we have access to $x_{v_l}$ since $x_{v_l}$ is from the training set. Now, each candidate $v_l$ induces a uniform distribution $\nu_{v_l}$ over its selected feature-time pairs (and termination), defined as $\nu_{v_l}(m, t') = \frac{1}{|v_l|}$ if $(m, t') \in v_l$ and 0 otherwise. Intuitively, $\nu_{v_l}$ spreads equal probability mass over all acquisitions in the future candidate plan $v_l$. In the case that $v_l$ contains no further acquisition, $\nu_{v_l}(\varnothing) = 1$. The future-candidate distribution $\zeta_o$ for observed features $x_o$ is the weighted sum of $\nu_{v_l}$ as

$$\zeta_o = \sum_{l=1}^{\kappa} w_l \, \nu_{v_l} \tag{7}$$

where $w_l = \frac{\exp(-r_l)}{\sum_{k=1}^{\kappa} \exp(-r_k)}$. Intuitively, given a partially observed feature $x_o$, $\zeta_o$ indicates how likely each feature at each timestep (and the termination) is chosen by weighting their accuracy-cost tradeoff. For each training sample $(x_i, y_i) \in \mathcal{D}$, we construct $\zeta_{i,o}$ accordingly.

### 3.4 NOCTA-P: PARAMETRIC METHOD

Above, we proposed a non-parametric, neighbor-based approach to estimate our generalized NOCTA cost objective Eq. (1). We also develop a direct parametric approach, which estimates the cost objective using a network. We expound on our parametric approach, coined NOCTA-P, below.

**Estimating** $u(x_o, o)$**.** In order to estimate $u(x_o, o)$ in Eq. (1) parametrically, we attempt to calculate the utility for all potential subsets $v \subseteq \mathcal{V}$ at current time $t$:

$$I_v(x_o, o, t) = \ell(x_o \cup x_v, y, t) + \alpha \sum_{(m, t') \in v} c^m. \tag{8}$$

Since we have access to data during training, this value can be directly calculated for all subsets $v$ for training instances $(x_i, y_i)$. However, this is not the case during evaluation time when $x_v$ becomes unknown. Therefore, we estimate $I_v(x_o, o, t)$ through a value network, which we describe below.

**Value Network.** The value network $f_\theta$ predict the utility $I_v(x_o, o, t)$ for all potential subsets. Since future features $x_v$ are unknown during evaluation, the network is only conditioned on observed

features $x_o$, a candidate mask $v$, and the current time $t$; and its output is denoted as $f_\theta(x_o, v, t)$. Since this output estimates the utility of $I_v(x_o, o, t)$, we optimize through the mean squared error as:

$$\mathcal{L}_{\text{value}} = \text{MSE}\left(f_\theta(x_o, v, t), I_v(x_o, o, t)\right) . \tag{9}$$

Using this value network, we can estimate Eq. (1) to derive the future acquisition plan $u(x_o, o)$.

To minimize the distribution shift between the features evaluated by the value network and those used by the predictor, we jointly optimize the prediction network $\hat{y}_k$ and the value network $f_\theta$ over the candidate subset, resulting in the final loss function as:

$$\mathcal{L}_{\text{NOCTA-P}} = \frac{1}{L} \sum_{k=1}^{L} \text{CE}\left(\hat{y}_k(x_{o_{1:k}}), y_k\right) + \lambda \cdot \mathcal{L}_{\text{value}} , \tag{10}$$

where $\lambda$ balances the tradeoff between minimizing the prediction error (the cross-entropy loss) and accurately estimating the utility of future acquisitions with the value network ($\mathcal{L}_{\text{value}}$).

## 4 EXPERIMENTS

### 4.1 DATASETS

**Synthetic.** Inspired by Kossen et al. (2022), we construct a synthetic dataset of $N = 8,000$ samples with $L = 10$ non-i.i.d timepoints. Each timepoint contains two features: *digit* and *counter*. Labels are the cumulative sum of the digit, where the counter values are zero. We expound below.

The *digit* feature is a sequence of uniformly random $L$ integers from $\{0, 1, 2\}$, e.g., 1010220110. The *counter* feature is created by concatenating countdown sequences that begin with randomly chosen starting values from $\{0, 1, 2\}$ and truncating to maintain a fixed length of $L$. For example, starting values 2, 1, 2, and 1 yield 210, 10, 210, 10, and the final counter feature is their concatenation 2101021010 (of length $L$). Unlike Kossen et al. (2022), we assign labels at every timestep: $y_t$ is the cumulative sum of the digit values at all previous timesteps where the corresponding counter value is 0, e.g., label 0011333444 in the case of $\genfrac{}{}{0pt}{}{\text{counter: } 2101021010}{\text{digit: } 1010220110}$. The policy should begin with acquiring the digit and counter at $t = 1$, then acquiring the digit at the anticipated 0 counter location, then the next counter, and so on. Following Kossen et al. (2022), we split data into train/val/test (70/15/15).

**ADNI.** The Alzheimer's Disease Neuroimaging Initiative (ADNI) dataset[2] (Petersen et al., 2010) is a longitudinal, multi-center, observational study. Patients are categorized into cognitively unimpaired, mild cognitive impairment, and Alzheimer's Disease. Following Qin et al. (2024), we use $N = 1,002$ patients with four biomarkers extracted from PET (FDG and AV45) and MRI (Hippocampus and Entorhinal) across $L = 12$ visits, and split the dataset into train/val/test (64/16/20).

**OAI.** The Osteoarthritis Initiative (OAI) dataset[3] contains $N = 4,796$ patients with separate left/right knee evaluations, and each patient is monitored longitudinally for up to 96 months. We use the tabular data from Chen et al. (2024) and joint space width, totaling 27 features per visit across $L = 7$ visits. We predict the two clinical scores: Kellgren-Lawrence grade (KLG) (Kellgren et al., 1957) (range $0 \sim 4$) and WOMAC pain score (McConnell et al., 2001) (range $0 \sim 20$). Following previous works (Chen et al., 2024; Nguyen et al., 2024), we merge KLG $= 0$ and 1, and define WOMAC $< 5$ as no pain and $\geq 5$ as pain, and split the dataset into train/val/test (50/12.5/37.5).

### 4.2 BASELINES

*We underline expand the comparisons performed in recent work* by Qin et al. (2024) in longitudinal AFA by comparing to *two other state-of-the-art AFA methods*, DIME (Gadgil et al., 2024) and DiFA (Ghosh and Lan, 2023), in addition to the RL method considered by Qin et al. (2024).

In total, we consider the following baselines: 1) **ASAC** (Yoon et al., 2019): actor-critic method to jointly train a feature selector and a predictor network; 2) **RAS** (Qin et al., 2024): RL to decide when and which features to capture for longitudinal data where the predicted acquisition times are contin-

---

[2]https://adni.loni.usc.edu
[3]https://nda.nih.gov/oai/

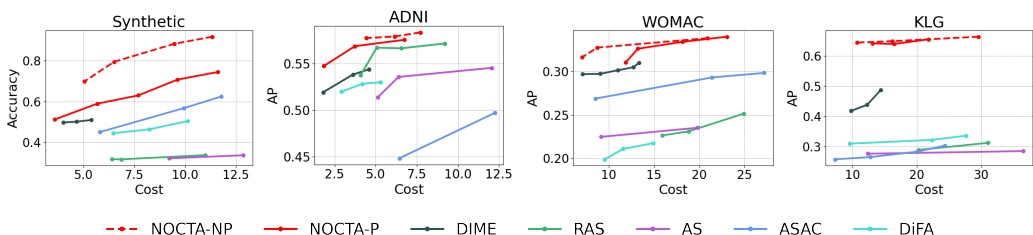

Figure 3: Performance/cost of models across various average acquisition costs (budgets). Following Kossen et al. (2022), we show accuracy rather than average precision for the synthetic dataset. Our `NOCTA` variants show the highest accuracies for a given cost. Best viewed digitally.

uous and not restricted to certain discrete timepoints; 3) **AS** (Qin et al., 2024): RAS with a constant acquisition interval; 4) **DIME** (Gadgil et al., 2024): greedy method to sequentially select the most informative features; 5) **DiFA** (Ghosh and Lan, 2023): Gumbel-softmax-based differentiable policy.

We adapt DIME and DiFA for longitudinal data by restricting acquisitions to either the current or future time points. Note that DIME and DiFA acquire features one at a time, allowing repeated selection within the same time step $t$, while `NOCTA` and other baselines simultaneously select all desired features at $t$ before moving to the next time steps. This gives both DiFA and DIME an advantage: they can decide whether to keep acquiring at the current visit or move forward.

### 4.3 IMPLEMENTATION

**Data Availability and Missingness.** Unlike standard AFA methods that assume fully observed data (Rahbar et al., 2025; Shim et al., 2018; Nan et al., 2015), our work is general and handles missingness completely at random. This is done by excluding missing features from candidate sets and pretraining the value and prediction network with random dropout of the input features.

**Prediction Network Implementation.** At each time $t$, the prediction network $\hat{y}_k$ predicts outcomes at future timepoints $k \geq t$. We train an MLP that takes all the features observed up to time t, denoted $x_{o_{1:t}}$ [4] and the future time indicator $k$. For predictions at $k > t$, the network must predict without access to any additional features beyond those observed by time $t$ (i.e., $\hat{y}_k \sim P(y_k|x_{o_{1:t}})$ for $k > t$).

**`NOCTA-NP` Implementation.** We pretrain the MLP predictor $\hat{y}_k$ and then train the MLP embedding network $g_\phi$, both optimized with Adam (Kingma, 2014) (learning rate of 1e−3). We set $\beta = \gamma = 1$, $\kappa = K = 5$, and embedding dimension $E = 32$ (Synthetic/ADNI) and $E = 64$ (OAI).

**`NOCTA-P` Implementation.** We first pretrain the MLP predictor $\hat{y}_k$, followed by jointly fine-tuning the predictor and the MLP-based value network $f_\theta$. Both steps use Adam (Kingma, 2014) with learning rate of $1e-3$ and use $\lambda = 1$.

**Subset Size $|\mathcal{O}|$.** For experiments, we uniformly sample 1000 candidate acquisitions to balance effectiveness and overhead. As shown in Appx. $A.6$, accuracy stabilizes and returns diminish beyond this cardinality. In practice, we observed uniform sampling is effective and fast, but other discrete optimization methods can be utilized (Parker and Rardin, 2014; Rajeev and Krishnamoorthy, 1992).

We report predictive performance results as the mean over five independent runs (standard errors for each run are provided in Appx. C). Feature costs and further details on the experimental setup can be found in Appx. B. Moreover, Appx. A.6 shows a hyperparameter sensitivity analysis, demonstrating that *our method remains robust across different settings and can be applied without extensive tuning*. Upon publication, we will make our code publicly available.

### 4.4 PERFORMANCE-COST RESULTS

We show results across different datasets in Fig. 3. For each figure, the x-axis represents different average acquisition costs (across instances), and the y-axis represents the performance. As the

---

[4]$x_o$ may contain missingness inherent to the data collection process. Additionally, unobserved features (missing or not acquired) are being masked out during implementation.

Table 1: Result comparisons of using the feature values, $\hat{y}$ prediction embedding (last hidden layer), and learned representation $g_\phi(\cdot)$ to compute nearest neighbors on datasets. We report the mean $\pm$ standard errors across five runs

| METHOD | ADNI | | | WOMAC | | | KLG | | |
|---|---|---|---|---|---|---|---|---|---|
| | AP | ROC | COST | AP | ROC | COST | AP | ROC | COST |
| FEATURE VALUES | $0.545 \pm 0.004$ | $0.727 \pm 0.002$ | $7.052 \pm 0.069$ | $0.315 \pm 0.003$ | $0.617 \pm 0.004$ | $9.014 \pm 2.863$ | $0.629 \pm 0.003$ | $0.821 \pm 0.002$ | $11.024 \pm 0.011$ |
| | $0.550 \pm 0.007$ | $0.732 \pm 0.008$ | $10.297 \pm 1.168$ | $0.331 \pm 0.013$ | $0.633 \pm 0.013$ | $17.574 \pm 0.070$ | $\mathbf{0.651 \pm 0.010}$ | $0.826 \pm 0.021$ | $17.741 \pm 0.033$ |
| PREDICTION EMBEDDING | $0.567 \pm 0.005$ | $0.744 \pm 0.004$ | $6.218 \pm 0.038$ | $0.290 \pm 0.005$ | $0.618 \pm 0.002$ | $6.998 \pm 0.023$ | $0.639 \pm 0.004$ | $\mathbf{0.825 \pm 0.002}$ | $13.706 \pm 0.055$ |
| | $0.571 \pm 0.021$ | $0.744 \pm 0.013$ | $7.866 \pm 0.068$ | $0.290 \pm 0.004$ | $0.618 \pm 0.002$ | $19.681 \pm 0.081$ | $0.648 \pm 0.001$ | $\mathbf{0.829 \pm 0.001}$ | $24.705 \pm 0.043$ |
| EMBEDDING NETWORK | $\mathbf{0.578 \pm 0.004}$ | $\mathbf{0.760 \pm 0.003}$ | $4.434 \pm 0.042$ | $\mathbf{0.328 \pm 0.004}$ | $\mathbf{0.646 \pm 0.003}$ | $8.729 \pm 0.011$ | $\mathbf{0.644 \pm 0.003}$ | $0.822 \pm 0.001$ | $10.687 \pm 0.018$ |
| | $\mathbf{0.584 \pm 0.013}$ | $\mathbf{0.760 \pm 0.006}$ | $7.727 \pm 0.094$ | $\mathbf{0.339 \pm 0.008}$ | $\mathbf{0.647 \pm 0.003}$ | $20.921 \pm 0.058$ | $0.649 \pm 0.002$ | $0.822 \pm 0.002$ | $16.192 \pm 0.062$ |

budget increases, more features can be acquired, and prediction performance typically improves. We use different hyperparameters to obtain prediction results under different average acquisition costs. Details on the performance-cost tradeoff hyperparameters are in Appx. B.

**Synthetic Dataset.** Following Kossen et al. (2022), we use accuracy for performance on the synthetic dataset. NOCTA-NP outperforms the other baselines, followed by NOCTA-P, indicating that the baseline methods fail to model the dependency between features and necessary acquisitions.

**ADNI Dataset.** Following Qin et al. (2024), we show the average precision (AP) result for different costs. We can see that both the parametric and non-parametric versions of NOCTA consistently outperform all other baseline methods while using lower overall acquisition cost.

**OAI Dataset.** Similar to the ADNI results, we show the AP result on KLG and WOMAC prediction in Fig. 3. NOCTA variants perform well and outperform the baselines, while RL-based approaches (ASAC, AS, and RAS) underperform compared to the greedy approach (DIME), suggesting that RL-based frameworks may struggle with the complexity and noise of longitudinal feature acquisition.

Overall, our NOCTA-NP model achieves highest accuracy across benchmarks, and our NOCTA-P model achieves comparable performance without the need to search for neighbors. For detailed time complexity and runtime analysis, please see Appx. A.4, and for additional results reporting AUC ROC, please refer to Appx. C.

## 4.5 ABLATION OF NOCTA-NP NEIGHBOR DISTANCE

We evaluate the effectiveness of the learned representation during the nearest neighbors process. That is, we compare defining neighbors using the learned representation $g_\phi$ versus using the native feature values and using the last hidden layer from the predictor $\hat{y}$. For any two samples $x_i$ and $x_j$ having the same observed feature $o$, we define feature-based distance as: $d(x_{i,o}, x_{j,o}) = \|x_{i,o} - x_{j,o}\|_2$. Moreover, we define the prediction embedding using the last hidden layer of the predictor $\hat{y}$, denoted as $h(\cdot)$. Thus, the distance for the prediction embedding is computed as $d(x_{i,o}, x_{j,o}) = \|h(x_{i,o}) - h(x_{j,o})\|_2$. Tab. 1 shows results under two cost regimes, low and high. Our non-parametric approach is robust to various choices of metrics and performs relatively well throughout. Moreover, the learned embedding $g_\phi(\cdot)$ outperforms both the feature values and the prediction embedding $h(\cdot)$ on both ADNI and WOMAC datasets, and it shows comparable results on the KLG dataset. Notably, on ADNI, the low-cost result for $g_\phi(\cdot)$ (cost = 4.343) already outperforms the high-cost performance of both alternatives (cost = 7.866 and 10.297). This indicates *the embedding network is able to learn informative representations* that can be used for real-world AFA tasks.

## 5 CONCLUSION

In this work, we proposed NOCTA, a non-greedy longitudinal active feature acquisition framework designed to directly balance acquisition cost with prediction accuracy. We introduced a cohesive estimation target for our NOCTA framework with two complementary solution strategies. Results on both synthetic and real-world medical datasets show that NOCTA outperforms other baseline methods while achieving lower overall acquisition cost. While our non-parametric approach provides consistently strong performance, the parametric approach provides comparable performance with more lightweight (non-neighbor based) inference. Either NOCTA-P or NOCTA-NP can be easily validated against held-out data or ensembles if one seeks a single policy.

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

APPENDIX

# A  ADDITIONAL DISCUSSION

## A.1  USAGE OF LLMS STATEMENT

We used Grammarly (https://www.grammarly.com/) to identify and correct grammatical errors. All revisions were manually reviewed by the authors to ensure the original meaning remained unchanged.

## A.2  BROADER IMPACT

In this paper, we introduce a non-greedy longitudinal active feature acquisition method, NOCTA, which could aid the decision-making process in real-world applications such as healthcare by selectively acquiring essential examinations; thus minimizing patient burden (e.g., time and costs) through personalized and timely intervention, as in any ML system, using NOCTA should involve thorough/robust evaluations and supervision by medical professionals to prevent any potential negative impact.

**Broader Application.** We acknowledge that our experiments focus on the healthcare section, an important application area; however, our method is not specific to any domain. Similar resource-constrained scenarios also exist in other domains, such as in energy-aware audio recognition (Monjur et al., 2023) or IoT devices (Dunkels et al., 2011). In fact, our AFA formulation has applications across a myriad of domains, including robotics (acquisition of sensors with resource constraints), education (acquisition of different exam types without over-burdening students), and beyond.

## A.3  ASSUMPTIONS & LIMITATIONS

Like traditional AFA methods and RL-based baselines, NOCTA assumes that the problem has a finite horizon with a limited number of time steps. Additionally, we acknowledge that, similar to other baselines presented in this paper, the performance of NOCTA is sensitive to the data quality. In particular, poor label annotation in real-world clinical datasets can degrade the accuracy. However, extending to an infinite-horizon setting and mitigating the sensitivity to the data quality are beyond the scope of this paper. We will leave these challenges for future work.

## A.4  TIME COMPLEXITY

With a total of time steps $L$ and a total number of features $M$, the time complexity is $O(ML^2)$ for RL baselines (RAS, AS, ASAC), and $O(M^2L^2)$ for DiFA and DIME. For our parametric method NOCTA-P, with a random subset search of size $|\mathcal{O}|$, the time complexity is $O(ML^2|\mathcal{O}|)$. However, if you want to incorporate the nearest neighbor NOCTA-NP with the number of training points $n$, the time complexity is $O(nML^2|\mathcal{O}|)$, where a naive nearest neighbor search is $O(nML)$. Note that NOCTA can further parallelize the $|\mathcal{O}|$ subsets across independent workers.

## A.5  PROOF OF THEOREM 1

**Theorem 1.** *(Informal)* NOCTA *lower bounds the value of the optimal longitudinal AFA MDP policy*

Problem Setup. For any observations $(x_o, o, t)$, let the set of *all* feature-time pairs that are available be:

$$\mathcal{V}_t = \{(m, t') \mid m \in \{1, \ldots, M\}, t' \in \{t+1, \ldots, L\}\}.$$

To follow the convention of reinforcement learning literature, we redefine the objective for NOCTA (Eq. (1)) as maximization instead of minimization without loss of generality. That is, for any partial observation $(x_o, t)$, we define $\text{NOCTA}_t^s(x_o)$ as:

$$\text{NOCTA}_t^s(x_o) := \max_{\substack{v \subseteq \mathcal{V}_t \\ \text{s.t } |\text{rounds}(v)| \leq s}} \left[ -\mathbb{E}_{y_{t+1:L}, x_v | x_o}[\ell(x_o \cup x_v, y, t)] - \alpha \sum_{(m, t') \in v} c^m \right],$$

where $s \in \mathbb{N}$ represents the maximum number of possible future acquisition rounds, and let $\text{rounds}(v) := \{t' \mid \exists m \in \{1, \ldots, M\} : (m, t') \in v\}$. We then define

$$V_t^0(x_o) := -\mathbb{E}_{y_{t+1:L}|x_o}[\ell(x_o, y, t)]$$

and

$$V_t^s(x_o) := \max\left\{V_t^0(x_o), \max_{\substack{t' \in \{t+1, \ldots, L\} \\ a_{t'} \in \{0,1\}^M}}\left[-\alpha \sum_m a_{t',m} c^m + \mathbb{E}_{x_{a_{t'}}|x_o}[V_{t'}^{s-1}(x_{o \cup a_{t'}})]\right]\right\}. \tag{11}$$

We see that $V_t^0$ is the expected negative cumulative loss when we immediately terminate at $t$, and we predict the future labels using only what we have acquired so far, $x_o$. Thus, $V_t^s$ is the value for the optimal longitudinal AFA policy, such that it either (i) terminates, returning $V_t^0$ or (ii) executes another acquisition at the future time step $t' \in (t+1, ..., L)$ and acquires the feature $a_{t'} \in \{0,1\}^M$.

Claim. For every $(x_o, t)$ with $s \in \mathbb{N} \geq 0$, we have $V_t^s(x_o) \geq \text{NOCTA}_t^s(x_o)$.

Proof.

*Base case $s = 0$.* Since there is no further action, both the MDP agent and NOCTA terminate:

$$V_t^0(x_o) = -\mathbb{E}_{y_{t+1:L}|x_o}[\ell(x_o, y, t)] = \text{NOCTA}_t^0(x_o).$$

*Inductive hypothesis.* Assume for $s - 1 \geq 0$, we have:

$$V_{t'}^{s-1}(x) \geq \text{NOCTA}_{t'}^{s-1}(x) \quad \forall(x, t'). \tag{IH}$$

*Inductive step.* We fix $(x_o, t)$ and see that:

$V_t^s(x_o)$

$$= \max\left\{V_t^0(x_o), \max_{\substack{t' \in \{t+1, \ldots, L\} \\ a_{t'} \in \{0,1\}^M}}\left[-\alpha \sum_m a_{t',m} c^m + \mathbb{E}_{x_{a'_t}|x_o}[V_{t'}^{s-1}(x_{o \cup a_{t'}})]\right]\right\}$$

$$\overset{\text{(IH)}}{\geq} \max\left\{V_t^0(x_o), \max_{\substack{t' \in \{t+1, \ldots, L\} \\ a_{t'} \in \{0,1\}^M}}\left[-\alpha \sum_m a_{t',m} c^m + \mathbb{E}_{x_{a'_t}|x_o}[\text{NOCTA}_{t'}^{s-1}(x_{o \cup a_{t'}})]\right]\right\}$$

$$= \max\Big\{V_t^0(x_o),$$

$$\max_{\substack{t' \in \{t+1, \ldots, L\} \\ a_{t'} \in \{0,1\}^M}}\left[-\alpha \sum_m a_{t',m} c^m + \mathbb{E}_{x_{a_{t'}}|x_o}\Big[\max_{\substack{v' \subseteq \mathcal{V}_{t'} \\ \text{s.t } |\text{rounds}(v')| \leq s-1}}\Big(-\mathbb{E}_{y_{t'+1:L}, x_{v'}|x_o, x_{a_{t'}}}[\ell(x_o \cup x_{a_{t'}} \cup x_{v'}, y, t')] - \alpha \sum_{(m,t'') \in v'} c^m\Big)\Big]\right]\Big\}$$

$$\overset{*}{\geq} \max\Big\{V_t^0(x_o),$$

$$\max_{\substack{t' \in \{t+1, \ldots, L\} \\ a_{t'} \in \{0,1\}^M}} \max_{\substack{v' \subseteq \mathcal{V}_{t'} \\ \text{s.t } |\text{rounds}(v')| \leq s-1}}\left[-\mathbb{E}_{y_{t'+1:L}, x_{v'}, x_{a_{t'}}|x_o}[\ell(x_o \cup x_{a_{t'}} \cup x_{v'}, y, t')] - \alpha\Big(\sum_{(m,t'') \in v'} c^m + \sum_m a_{t',m} c^m\Big)\right]\Big\}$$

$$= \max\Big\{V_t^0(x_o), \max_{\substack{v \in \mathcal{V}_t \\ \text{s.t } |\text{rounds}(v) \leq s|}}[-\mathbb{E}_{y_{t+1:L}, x_v|x_o}[\ell(x_o \cup x_v, y, t)] - \alpha \sum_{(m,t'') \in v} c^m]\Big\}$$

$$= \text{NOCTA}_t^s(x_o).$$

where * follows from:

$$\forall v' \in \Omega, x_{a_{t'}}, \max_{v \in \Omega} \mathcal{L}\left(x_{a_{t'}}, v\right) \geq \mathcal{L}\left(x_{a_{t'}}, v'\right) \implies$$

$$\forall v' \in \Omega, \mathbb{E}_{x_{a_{t'}}|x_o}\left[\max_{v \in \Omega} \mathcal{L}\left(x_{a_{t'}}, v\right)\right] \geq \mathbb{E}_{x_{a_{t'}}|x_o}\left[\mathcal{L}\left(x_{a_{t'}}, v'\right)\right] \implies$$

$$\mathbb{E}_{x_{a_{t'}}|x_o}\left[\max_{v \in \Omega} \mathcal{L}\left(x_{a_{t'}}, v\right)\right] \geq \max_{v \in \Omega} \mathbb{E}_{x_{a_{t'}}|x_o}\left[\mathcal{L}\left(x_{a_{t'}}, v\right)\right]$$

for $\Omega := \left\{ v \subseteq \mathcal{V}_{t'} \;\middle|\; |\text{rounds}(v)| \leq s - 1 \right\}$ and

$$\mathcal{L}(x_{a_{t'}}, v) := -\mathbb{E}_{y_{t+1:L}, x_{v'} | x_o, x_{a_{t'}}} [\ell(x_o \cup x_{a_{t'}} \cup x_{v'}, y, t)] - \alpha \left( \sum_{(m, t'') \in v'} c^m + \sum_m a_{t', m} c^m \right).$$

Note that one may recover the non-cardinality constrained problem by considering large enough $s$, yielding a lower bound on the longitudinal AFA MDP.

## A.6 Sensitivity Analysis

In this section, we provide further explanations for hyperparameters and how the hyperparameters affect performance. Overall, we find that our proposed method is robust to different settings, indicating that our method can be applied readily without excessive fine-tuning.

$|\mathcal{O}|$: this specifies the size of the candidate subset in our search. As $|\mathcal{O}|$ increases, NOCTA can explore a broader set of potential acquisition plans. As seen in Table 2 and 3, both NOCTA-NP and NOCTA-P exhibit clear diminishing returns once $|\mathcal{O}|$ exceeds 1000.

$\kappa$: this controls how many top candidate subsets are aggregated to form the future acquisition distribution. Please refer to Table 5 for different settings.

$K$: this controls how many neighbors in the learned embedding space are used to estimate future acquisition gains. Small values of $K$ might result in high variance from too few neighbors, while high $K$ may include dissimilar samples. Please refer to Table 4 for different settings.

$\beta$: this controls the sensitivity that translates Jessen-Shannon divergence into a similarity score. A larger $\beta$ will penalize divergences more strongly, while a smaller $\beta$ will be more lenient. Please refer to Table 6 for different settings.

$\gamma$: this determines the margin in the contrastive loss, making sure that dissimilar embeddings are pushed far apart. If $\gamma$ is too small, the embedding network might not push dissimilar pairs far enough, and if $\gamma$ is too large, it might push moderately different samples by a large distance. Please refer to Table 7 for different settings.

Table 2: Ablation on the subset size $|\mathcal{O}|$ for NOCTA-NP used in the synthetic dataset

| $|\mathcal{O}|$ | Accuracy | Cost |
|---|---|---|
| 100 | $0.907 \pm 0.001$ | $13.252 \pm 0.044$ |
| 1000 | $0.919 \pm 0.002$ | $11.345 \pm 0.012$ |
| 5000 | $0.921 \pm 0.001$ | $12.046 \pm 0.021$ |
| 10000 | $0.921 \pm 0.002$ | $11.945 \pm 0.012$ |

Table 3: Ablation on the subset size $|\mathcal{O}|$ for NOCTA-P used in the synthetic dataset

| $|\mathcal{O}|$ | Accuracy | Cost |
|---|---|---|
| 100 | $0.731 \pm 0.010$ | $11.401 \pm 0.040$ |
| 1000 | $0.745 \pm 0.006$ | $11.624 \pm 0.087$ |
| 5000 | $0.749 \pm 0.016$ | $11.430 \pm 0.115$ |
| 10000 | $0.742 \pm 0.017$ | $11.419 \pm 0.150$ |

Table 4: Ablation on the number of neighbors $K$ used in the synthetic dataset

| $K$ | Accuracy | Cost |
|---|---|---|
| 5 | $0.888 \pm 0.005$ | $9.471 \pm 0.033$ |
| 10 | $0.874 \pm 0.003$ | $9.489 \pm 0.006$ |
| 25 | $0.889 \pm 0.001$ | $9.512 \pm 0.006$ |

Table 5: Ablation on the number of top candidate subset $\kappa$ used in the synthetic dataset

| $\kappa$ | Accuracy | Cost |
|---|---|---|
| 5 | $0.884 \pm 0.005$ | $9.471 \pm 0.033$ |
| 10 | $0.883 \pm 0.001$ | $9.618 \pm 0.022$ |
| 25 | $0.880 \pm 0.002$ | $9.564 \pm 0.067$ |

Table 7: Ablation on the margin in contrastive loss $\gamma$ used in the synthetic dataset

| $\gamma$ | Accuracy | Cost |
|---|---|---|
| 0.5 | $0.882 \pm 0.010$ | $9.592 \pm 0.035$ |
| 1 | $0.884 \pm 0.005$ | $9.471 \pm 0.033$ |
| 2 | $0.879 \pm 0.005$ | $9.638 \pm 0.033$ |
| 5 | $0.880 \pm 0.001$ | $9.688 \pm 0.017$ |

## B EXPERIMENTAL SETUP

### B.1 COST OF FEATURES

**Synthetic data.** Our synthetic dataset includes 2 features, one being the digit and the other being the counter. Since both are generated through random selection, we assign an equal cost, 1, to the two features. Upon publication, we will release the synthetic dataset.

**ADNI.** The ADNI dataset includes 4 features per time point, with FDG and AV45 extracted from PET scan and Hippocampus and Entorhinal extracted from MRI scan. Since PET is a more expensive diagnosis, we follow Qin et al. (2024) and assign a cost 1 to FDG and AV45, and a cost 0.5 to Hippocampus and Entorhinal. Access to ADNI data may be requested via `https://adni.loni.usc.edu`, and the Data Use Agreement is available at `https://adni.loni.usc.edu/terms-of-use/`.

**OAI.** We use a total of 27 features for the OAI dataset, with 17 clinical measurements, and 10 joint space width (JSW) extracted from knee radiography. For the clinical measurements, we assign a low cost of 0.3 to those that require minimum effort, e.g., age, sex, and race, and a slightly higher cost of 0.5 to blood pressure and BMI calculation. For JSW, a cost of 1.0 is assigned for the minimum JSW and 0.8 for those measured at different positions. Access to the OAI data may be requested via `https://nda.nih.gov/oai/`.

### B.2 BASELINE IMPLEMENTATION

Following Qin et al. (2024), we share the same neural CDE predictor (Kidger et al., 2020) for ASAC, RAS, and AS. For ASAC, RAS, and AS, we use the implementation available at `https://github.com/yvchao/cvar_sensing`, which is under the BSD-3-Clause license. For DIME (Gadgil et al., 2024), we extend the method to longitudinal by building on top of the official implementation available at `https://github.com/suinleelab/DIME`. The DiFA codebase is not publicly released, and researchers may request it directly from the authors.

**ASAC (Yoon et al., 2019).** We select the coefficient for the acquisition cost: 1) $\mu \in \{0.0015, 0.002, 0.003\}$ for the synthetic dataset, 2) $\mu \in \{0.005, 0.02\}$ for ADNI,

Table 6: Ablation on the scaling in similarity function $\beta$ used in the synthetic dataset

| $\beta$ | Accuracy | Cost |
|---|---|---|
| 0.5 | $0.879 \pm 0.002$ | $9.660 \pm 0.054$ |
| 1 | $0.884 \pm 0.005$ | $9.471 \pm 0.033$ |
| 2 | $0.890 \pm 0.001$ | $9.592 \pm 0.022$ |
| 5 | $0.888 \pm 0.003$ | $9.640 \pm 0.048$ |

3) $\mu \in \{0.0012, 0.0016, 0.003\}$ for the WOMAC of OAI dataset, and 4) $\mu \in \{0.00125, 0.0015, 0.00175, 0.005\}$ for the KLG of OAI dataset.

**RAS (Qin et al., 2024).** For the synthetic dataset, we set the allowed acquisition interval to $(\Delta_{\min} = 0.2, \Delta_{\max} = 1.0)$ for the synthetic dataset and $(\Delta_{\min} = 0.5, \Delta_{\max} = 4.5)$ for ADNI, WOMAC, and KLG.

Moreover, we use the coefficient for diagnostic error: 1) $\gamma \in \{5000, 7000, 9000\}$ for the synthetic dataset, 2) $\gamma \in \{50, 75, 100, 175\}$ for ADNI dataset, 3) $\gamma \in \{2000, 4500, 9000\}$ for WOMAC of OAI, and 4) $\gamma \in \{1200, 1700\}$ for KLG of OAI dataset. For all datasets, we use a tail-risk quantile of $0.1$, a penalty for invalid visits of $10$, and a discount factor (for tackling long trajectories) of $0.99$.

**AS (Qin et al., 2024).** We set the minimum and maximum allowed acquisition intervals as in the RAS setting. In addition, we select fixed acquisition interval of: 1) $\tilde{\Delta} \in \{0.2, 0.4\}$ for the synthetic dataset, 2) $\tilde{\Delta} \in \{0.5, 1.0, 1.5\}$ for ADNI dataset, 3) $\tilde{\Delta} \in \{0.5, 1.5\}$ for WOMAC of OAI, and 4) $\tilde{\Delta} \in \{0.5, 1.5\}$ for KLG of OAI dataset.

**DIME (Gadgil et al., 2024).** For this baseline, we extend the original model to the longitudinal setting, while still keeping its greedy feature acquisition strategy. Both the prediction network and value network share the same architecture as our `NOCTA-P` model, and hence we use the same training strategies.

**DiFA (Ghosh and Lan, 2023).** Following Ghosh and Lan (2023), we use a variational autoencoder with arbitrary conditioning (VAEAC) probabilistic model (Ivanov et al., 2018) for imputation. When using the same architecture with `NOCTA-P`, DiFA performs poorly. Hence, we gave DiFA an advantage and used the recommended architecture from the DiFA paper. That is, both the prediction network and the feature policy model have two fully connected layers (hidden size 128) with skip connections, dropout regularizer, and LeakyReLU activation function.

### B.3 NOCTA IMPLEMENTATION

**Prediction Network.** We share the same MLP architecture for both `NOCTA-NP` and `NOCTA-P`. Specifically, the network consists of two hidden layers with ReLU. Each hidden layer contains 10 units for the synthetic and ADNI datasets, and 32 units for the WOMAC and KLG of the OAI dataset, respectively. Moreover, we apply random dropout for the input during training of the predictor.

**NOCTA-NP.** We select the accuracy-cost trade-off hyperparameter: 1) $\alpha \in \{0.002, 0.006, 0.02, 0.03\}$ for the synthetic dataset, 2) $\alpha \in \{0.003, 0.005, 0.01\}$ for ADNI dataset, 3) $\alpha \in \{0.00075, 0.005, 0.01\}$ for WOMAC of OAI dataset, and 4) $\alpha \in \{0.0025, 0.001, 0.006\}$ for KLG of OAI dataset.

For the embedding network, we use four hidden layers with ReLU. Each hidden layer contains 32 units for the synthetic and ADNI datasets, and 64 units for the WOMAC and KLG of the OAI dataset, respectively.

**NOCTA-P.** For the value network, we use an MLP architecture with two hidden layers and ReLU as the activation function. We use the same hidden size as the prediction network, but the number of outputs is set to be 1 for estimating the utility of the potential subset. We set the accuracy-cost trade-off hyperparameter $\alpha$ to $5e-5$ for WOMAC and $5e-4$ for all other tasks, and the acquisition terminates by either selecting the empty set or reaching the budget.

To promote more stable training for Eq. (8) in `NOCTA-P`, we slightly modify the value network objective in two ways: 1) remove the acquisition cost and train the value network to estimate only

the CE loss, i.e., $\bar{I}_v(x_o, o) = I_v(x_o, o) - \alpha \sum_{(m,t') \in v} c^m$. Since the cost of each subset is fixed and can be directly added during subset selection, excluding it from the prediction target simplifies the learning process without affecting the final decision; 2) consider the relative prediction improvement over the current feature set: $\bar{\bar{I}}_v(x_o, o) = \ell(x_o, y, t) - \bar{I}_v(x_o, o)$. This represents the potential gain from acquiring new features and implicitly normalizes the loss scale. Since $\ell(x_o, y, t)$ is a constant, this will not change the selection (i.e., through $\arg\max$), and hence remains the same objective as Eq. (1).

### B.4 HARDWARES

We employed two different clusters for our experiments: (1) Intel Xeon CPU E5-2630 v4 and NVIDIA GeForce GTX 1080 for experiments, and (2) Intel Xeon Silver 4114 CPU and NVIDIA GeForce GTX 2080. We used cluster (1) for RAS, AS, ASAC, DiFA, `NOCTA-NP`, and cluster (2) for DIME and `NOCTA-P`.

## C ADDITIONAL RESULTS

### C.1 ROC AUC ON REAL-WORLD DATASETS

Following Qin et al. (2024), Figure 4 shows additional ROC AUC results on real-world datasets across varying average acquisition budgets. Our method consistently outperforms all baselines while using a lower cost.

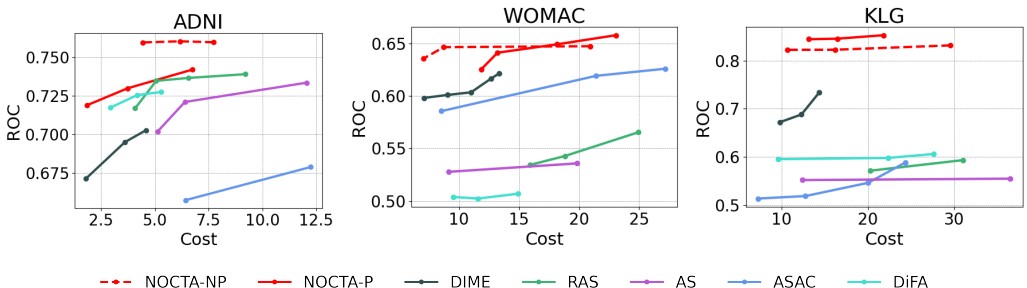

Figure 4: Additional results on performance/cost of models, measured by ROC AUC across varying average acquisition budgets on real-world datasets. Best viewed in color.

### C.2 FULL RESULTS WITH MEAN AND STANDARD DEVIATION

Tables 8, 9, 10, and 11 report the mean and standard deviation for each data point presented in Figures 3 and 4, computed over five independent runs.

Table 8: Full results for accuracy vs. average cost on Synthetic Dataset.

| METHOD | SYNTHETIC | |
|---|---|---|
| | ACCURACY | COST |
| ASAC | $0.449 \pm 0.078$ | $5.803 \pm 5.180$ |
| | $0.567 \pm 0.161$ | $9.929 \pm 8.365$ |
| | $0.624 \pm 0.136$ | $11.780 \pm 5.211$ |
| RAS | $0.316 \pm 0.031$ | $6.390 \pm 5.316$ |
| | $0.315 \pm 0.032$ | $6.852 \pm 5.129$ |
| | $0.336 \pm 0.022$ | $11.003 \pm 2.638$ |
| AS | $0.320 \pm 0.014$ | $9.210 \pm 2.344$ |
| | $0.335 \pm 0.033$ | $12.870 \pm 3.762$ |
| DIME | $0.497 \pm 0.078$ | $4.000 \pm 0.000$ |
| | $0.501 \pm 0.080$ | $4.662 \pm 0.557$ |
| | $0.509 \pm 0.084$ | $5.375 \pm 1.213$ |
| DIFA | $0.444 \pm 0.015$ | $6.466 \pm 0.058$ |
| | $0.463 \pm 0.017$ | $8.249 \pm 0.420$ |
| | $0.504 \pm 0.010$ | $10.138 \pm 0.129$ |
| NOCTA-NP | $0.698 \pm 0.008$ | $5.028 \pm 0.022$ |
| | $0.795 \pm 0.002$ | $6.504 \pm 0.005$ |
| | $0.884 \pm 0.005$ | $9.471 \pm 0.033$ |
| | $0.919 \pm 0.002$ | $11.345 \pm 0.012$ |
| NOCTA-P | $0.512 \pm 0.018$ | $3.580 \pm 0.292$ |
| | $0.589 \pm 0.022$ | $5.666 \pm 0.221$ |
| | $0.630 \pm 0.026$ | $7.697 \pm 0.174$ |
| | $0.707 \pm 0.017$ | $9.624 \pm 0.164$ |
| | $0.745 \pm 0.006$ | $11.624 \pm 0.087$ |

Table 9: Full results for AP and ROC vs. average cost on ADNI prediction.

| METHOD | ADNI | | |
|---|---|---|---|
| | AP | ROC | COST |
| ASAC | $0.449 \pm 0.041$ | $0.657 \pm 0.038$ | $6.442 \pm 6.093$ |
| | $0.497 \pm 0.034$ | $0.679 \pm 0.015$ | $12.242 \pm 2.648$ |
| RAS | $0.538 \pm 0.036$ | $0.717 \pm 0.024$ | $4.087 \pm 0.735$ |
| | $0.567 \pm 0.015$ | $0.735 \pm 0.008$ | $5.064 \pm 0.324$ |
| | $0.567 \pm 0.022$ | $0.737 \pm 0.012$ | $6.560 \pm 2.459$ |
| | $0.572 \pm 0.012$ | $0.739 \pm 0.011$ | $9.197 \pm 1.438$ |
| AS | $0.514 \pm 0.046$ | $0.702 \pm 0.038$ | $5.133 \pm 1.659$ |
| | $0.536 \pm 0.021$ | $0.721 \pm 0.014$ | $6.394 \pm 2.477$ |
| | $0.546 \pm 0.021$ | $0.734 \pm 0.013$ | $12.054 \pm 1.763$ |
| DIME | $0.519 \pm 0.037$ | $0.671 \pm 0.047$ | $1.801 \pm 0.050$ |
| | $0.538 \pm 0.045$ | $0.695 \pm 0.048$ | $3.607 \pm 0.084$ |
| | $0.544 \pm 0.048$ | $0.703 \pm 0.049$ | $4.587 \pm 0.330$ |
| DIFA | $0.520 \pm 0.027$ | $0.717 \pm 0.029$ | $2.925 \pm 0.127$ |
| | $0.529 \pm 0.025$ | $0.725 \pm 0.028$ | $4.180 \pm 0.043$ |
| | $0.530 \pm 0.027$ | $0.727 \pm 0.033$ | $5.292 \pm 0.107$ |
| NOCTA-NP | $0.578 \pm 0.004$ | $0.760 \pm 0.003$ | $4.434 \pm 0.042$ |
| | $0.579 \pm 0.009$ | $0.760 \pm 0.003$ | $6.164 \pm 0.068$ |
| | $0.584 \pm 0.013$ | $0.760 \pm 0.006$ | $7.727 \pm 0.094$ |
| NOCTA-P | $0.548 \pm 0.025$ | $0.719 \pm 0.025$ | $1.851 \pm 0.019$ |
| | $0.569 \pm 0.023$ | $0.730 \pm 0.015$ | $3.729 \pm 0.079$ |
| | $0.576 \pm 0.021$ | $0.742 \pm 0.012$ | $6.743 \pm 0.330$ |

Table 10: Full results for AP and ROC vs. average cost on WOMAC score prediction.

| METHOD | WOMAC | | |
|---|---|---|---|
| | AP | ROC | COST |
| ASAC | $0.269 \pm 0.032$ | $0.586 \pm 0.060$ | $8.541 \pm 2.136$ |
| | $0.293 \pm 0.018$ | $0.619 \pm 0.028$ | $21.423 \pm 6.480$ |
| | $0.299 \pm 0.025$ | $0.626 \pm 0.026$ | $27.175 \pm 6.933$ |
| RAS | $0.226 \pm 0.008$ | $0.534 \pm 0.019$ | $15.930 \pm 3.475$ |
| | $0.231 \pm 0.007$ | $0.543 \pm 0.009$ | $18.810 \pm 1.802$ |
| | $0.251 \pm 0.019$ | $0.565 \pm 0.024$ | $24.939 \pm 3.796$ |
| AS | $0.225 \pm 0.005$ | $0.528 \pm 0.008$ | $9.160 \pm 2.271$ |
| | $0.235 \pm 0.011$ | $0.536 \pm 0.012$ | $19.826 \pm 10.326$ |
| DIME | $0.297 \pm 0.025$ | $0.598 \pm 0.043$ | $7.073 \pm 0.093$ |
| | $0.297 \pm 0.025$ | $0.601 \pm 0.043$ | $9.036 \pm 0.198$ |
| | $0.302 \pm 0.028$ | $0.603 \pm 0.044$ | $11.009 \pm 0.310$ |
| | $0.305 \pm 0.027$ | $0.617 \pm 0.037$ | $12.696 \pm 0.357$ |
| | $0.310 \pm 0.028$ | $0.621 \pm 0.038$ | $13.325 \pm 0.643$ |
| DIFA | $0.199 \pm 0.009$ | $0.504 \pm 0.004$ | $9.532 \pm 0.091$ |
| | $0.211 \pm 0.005$ | $0.502 \pm 0.002$ | $11.584 \pm 0.630$ |
| | $0.217 \pm 0.004$ | $0.507 \pm 0.008$ | $14.936 \pm 1.433$ |
| NOCTA-NP | $0.317 \pm 0.003$ | $0.636 \pm 0.0012$ | $7.063 \pm 0.004$ |
| | $0.328 \pm 0.004$ | $0.646 \pm 0.003$ | $8.729 \pm 0.011$ |
| | $0.339 \pm 0.008$ | $0.647 \pm 0.003$ | $20.921 \pm 0.058$ |
| NOCTA-P | $0.311 \pm 0.007$ | $0.625 \pm 0.006$ | $11.862 \pm 0.074$ |
| | $0.327 \pm 0.020$ | $0.641 \pm 0.018$ | $13.212 \pm 0.180$ |
| | $0.335 \pm 0.015$ | $0.649 \pm 0.015$ | $18.178 \pm 0.082$ |
| | $0.340 \pm 0.013$ | $0.658 \pm 0.011$ | $23.061 \pm 0.059$ |

Table 11: Full results for AP and ROC vs. average cost on KLG prediction.

| METHOD | KLG | | |
|---|---|---|---|
| | AP | ROC | COST |
| ASAC | $0.258 \pm 0.021$ | $0.513 \pm 0.035$ | $7.284 \pm 4.694$ |
| | $0.266 \pm 0.018$ | $0.519 \pm 0.019$ | $12.756 \pm 6.584$ |
| | $0.283 \pm 0.009$ | $0.546 \pm 0.024$ | $20.025 \pm 6.504$ |
| | $0.304 \pm 0.008$ | $0.589 \pm 0.015$ | $24.361 \pm 15.940$ |
| RAS | $0.290 \pm 0.012$ | $0.571 \pm 0.023$ | $20.253 \pm 11.269$ |
| | $0.313 \pm 0.049$ | $0.593 \pm 0.063$ | $31.044 \pm 8.560$ |
| AS | $0.277 \pm 0.012$ | $0.552 \pm 0.021$ | $12.394 \pm 2.027$ |
| | $0.285 \pm 0.022$ | $0.555 \pm 0.015$ | $36.499 \pm 8.124$ |
| DIME | $0.418 \pm 0.016$ | $0.672 \pm 0.023$ | $9.787 \pm 0.040$ |
| | $0.438 \pm 0.018$ | $0.688 \pm 0.028$ | $12.281 \pm 0.005$ |
| | $0.486 \pm 0.026$ | $0.734 \pm 0.024$ | $14.359 \pm 0.148$ |
| DIFA | $0.310 \pm 0.006$ | $0.595 \pm 0.019$ | $9.611 \pm 1.743$ |
| | $0.322 \pm 0.011$ | $0.598 \pm 0.015$ | $22.331 \pm 3.465$ |
| | $0.336 \pm 0.010$ | $0.606 \pm 0.013$ | $27.615 \pm 6.411$ |
| NOCTA-NP | $0.644 \pm 0.003$ | $0.822 \pm 0.001$ | $10.687 \pm 0.018$ |
| | $0.649 \pm 0.002$ | $0.822 \pm 0.002$ | $16.192 \pm 0.062$ |
| | $0.663 \pm 0.004$ | $0.831 \pm 0.001$ | $29.546 \pm 0.052$ |
| NOCTA-P | $0.641 \pm 0.054$ | $0.844 \pm 0.020$ | $13.144 \pm 0.107$ |
| | $0.639 \pm 0.056$ | $0.845 \pm 0.022$ | $16.448 \pm 0.228$ |
| | $0.655 \pm 0.042$ | $0.853 \pm 0.017$ | $21.776 \pm 0.323$ |

