# OpenReview forum: "NOCTA: Non-Greedy Objective Cost-Tradeoff Acquisition for Longitudinal Data"
_ICLR.cc/2026/Conference — ICLR 2026 Conference Withdrawn Submission_

### Official Review · Reviewer_PtTU · 2025-10-31

**Soundness:** 2
**Presentation:** 3
**Contribution:** 2
**Rating:** 4
**Confidence:** 4

**Summary:**

This paper tackles the problem of longitudinal active feature acquisition (AFA)—a key challenge in settings such as healthcare, where diagnostic features (e.g., lab tests or imaging studies) must be selectively acquired over time. The objective is to learn a policy that determines which features to acquire and when, balancing predictive accuracy against cumulative acquisition cost. The authors motivate their approach by pointing out the shortcomings of existing paradigms: reinforcement learning (RL) methods often struggle with large, high-dimensional state–action spaces and complex credit assignment, while greedy strategies tend to be suboptimal because they ignore the joint benefit of future acquisitions.

Empirical evaluations on one synthetic dataset and two real-world medical datasets (ADNI and OAI) demonstrate that the proposed method consistently outperforms both RL-based and extension of static AFA baselines, achieving higher predictive accuracy or average precision for a given acquisition budget.

**Strengths:**

-	The paper addresses a challenging and high-impact problem.
-	The proposed non-RL objective (Eq. 1) offers an elegant and intuitive formulation that directly optimizes the trade-off between predictive accuracy and acquisition cost, presenting a compelling alternative to complex RL-based approaches.

**Weaknesses:**

A key benefit of greedy objectives—typically formulated as the immediate gain from acquiring the next feature—is their ability to drastically reduce the search space into $2^M$ possible subsets (i.e., a manageable sequential selection process at each step). Although greedy methods can produce myopic decisions, prior work (e.g., Qin et al., 2024) has addressed this limitation by incorporating discounted future rewards, allowing the model to anticipate the downstream effects of current actions and learn when to acquire new features. In contrast, the proposed objective in Equation (1) scales with both the number of features and time steps ($L\times 2^M$), making it computationally challenging to estimate reliable action values in high-dimensional settings. Furthermore, despite being described as a “non-greedy” formulation, the method remains locally greedy, as each acquisition decision is made without fully considering subsequent selections. The use of deterministic (non-stochastic) subset selection can potentially cause the model to miss globally optimal or even competitive joint acquisition strategies as the actions are not fully explored during training.

**Questions:**

- Policy gradient methods are known to be unstable. The relatively strong performance of AS and RAS on the ADNI dataset (which was also used in their original papers) but their substantial performance drop on the new datasets suggests that these baselines may not have been properly tuned. How were the hyperparameters for these methods optimized?

- While it is commendable that baselines such as DIME and DiFA were adapted to the longitudinal setting, how were their hyperparameters tuned to ensure a fair comparison with the proposed method?

- Does the deterministic selection of future acquisitions risk premature convergence or local optima? Would incorporating stochastic sampling or uncertainty-aware estimation improve exploration and robustness?

- Including a performance upper bound (i.e., the result when all features are observed) would help contextualize the reported results and better illustrate the potential performance gap.

---

### Official Review · Reviewer_f4Cs · 2025-11-01

**Soundness:** 2
**Presentation:** 1
**Contribution:** 2
**Rating:** 2
**Confidence:** 2

**Summary:**

Longitudinal Active Feature Acquisition (AFA) is the problem of selecting the time points and the subset of features to be acquired at each time point. This paper presents a new method for Longitudinal AFA called NOCTA which finds the feature-time pairs that minimize a cost-adjusted expected loss over future observations. When this objective is minimized by acquiring no features, the method stops and returns the predictions based on the data observed so far. To evaluate the expectation over future observations, there are two variants of NOCTA. The first variant, NOCTA-NP, uses training observations that are nearest neighbors (based on a learned embedding) to the features observed so far. The second variant, NOCTA-P, uses a trained value network to predict the utility of all potential subsets of features.

**Strengths:**

The paper considers an important problem that is common in clinical settings: when should we observe a patient, what should we observe when we do, and when should we stop observing? The proposed method, which is discussed in detail, shows better experimental performance compared to baselines on four datasets (three with real data). The approach differs from existing works that use RL, which is not well suited to the challenges of longitudinal AFA.

**Weaknesses:**

The details of the paper were at times difficult to follow. I understand that the nature of the problem necessitates many variables and subscripts, but there were parts I was unable to understand. I filled in the paper summary based on my best understanding, but I had trouble understanding details of the paper, especially in sections 3.3 and 3.4. For me there was a disconnect between the problem setup in the introduction, which made sense, and the details in the methodology section, which I found difficult to follow. Clarifying the questions may impact my final review.

**Questions:**

Questions:
- I am confused about the train and test setup. In line 193, it says “Because the full trajectories $(x_{\nu}, y_{t+1:L})$ are available for the training set $\mathcal{D}$, the expectation in Eq. (1) can be evaluated during training”. Later, in line 215 it says “However, evaluating this objective is intractable as in practice we do not have access to the true expectation of labels and unacquired values for the test samples, $E_{y_{t+1:L}, x_\nu \mid x_o}$ in Eq. (1)”. Is the second reference to test time? The notation does not clearly distinguish between train and test time. Are sections 3.3 and 3.4 about evaluating Eq (1) on a separate test dataset that wasn’t defined? I would wonder if the train dataset were what had been observed so far, but this is not consistent with the Notation section as I understand it (since each $x_i$ goes until the final time L). It would be very helpful if this could be clarified.
- Are the embedding and/or value network what is being trained during training time? I see the classifier $\hat{y}$ is pretrained, but in the implementation it says the authors pretrain it. How is it pretrained? The references to “train” and “pretrain” in this section get to my confusion about the setup.
- Relatedly, after Equation 3, are the observed features o in the test set? (I.e., so the nearest neighbors are in the train set)
- I’m confused about the notation of $x_i$. In the Notation section, it says “$x_{i,t} \in \mathbb{R}$ includes measurements of M (costly) features”. How can it include multiple measurements if it is a scalar? In the following sentence, the term “multi-dimensional feature vectors” is confusing to me. Does this mean vectors of dimension greater than 1 or a collection of vectors? I am working off the assumption that multiple features (at most M) can be acquired at each time point.
- When a feature is acquired at time $t$, is it acquired across all $N$ instances? (Or, related to the confusion above, is this data acquired for a new test set of instances?)
- Also relatedly, is Algorithm 1 about training or testing? I assume testing.
- The confusions above made it difficult to understand equations 6 and 8.

---

### Official Review · Reviewer_tice · 2025-11-01

**Soundness:** 3
**Presentation:** 3
**Contribution:** 2
**Rating:** 4
**Confidence:** 3

**Summary:**

This paper introduces longitudinal active feature acquisition (AFA), where an agent decides at each visit which costly features to acquire and when, while predicting outcomes at intermediate timepoints. It proposes a non-greedy objective that directly optimizes the accuracy–cost trade-off over future horizons with two estimators: NOCTA-NP, which uses neighbor-based prediction in a learned embedding space, and NOCTA-P, a value network that scores candidate acquisition plans. The method executes only the next action and replans after new observations and achieves SoTA performance, outperforming RL, greedy, and differentiable AFA baselines.

**Strengths:**

- I think the idea is quite relevant, since many recent works either consider non-temporal AFA or constrained AFA. Allowing to select tests again reflects how tests are done in practice
- The objective in (1)  and the RL-free optimization is sound

**Weaknesses:**

Weaknesses/Questions:
- The claim that $|\mathcal{O}| = 1000$ is sufficient, beyond which we get diminishing returns is a bit flawed in my opinion; in particular the combinatorial space of candidate plans grow as $2^{M(L-t)}$, and the ablation with M=2, is insufficient. It would be great if the authors can compare with growing feature sizes (maybe only work with subsets of OAI?)
- It is hard to interpret the pareto-optimal curves, as I would be more interested to see where the decision-making changes, w.r.t time-step. In particular, the plots show a very large gap between NOCTA and baselines. While I am not extremely familiar with the baselines, I would expect methods which sequentially select to choose good features, but that doesn't seem the case.
	- Is it possible to plot w.r.t time for each?
	- Another possibility is that the initial feature set chosen by baselines is quite bad. What happens if all the baselines are conditioned to start with the same feature set at $t=0$? This mimics real-world setting, where for example, if a patient checks into the hospital with a condition, a set of preliminary tests are immediately done. This would give a slightly fairer playground.
- Another interesting point would be to note if the feature sets chosen make practical sense (for eg. in the ADNI or OAI datasets). Do correlations exist between time steps for features chosen? Are there cheap individual features which are extremely good predictors and the other features are redundant? Since we aren't aware of the costs, other statistics of the feature acquisition is useful (#features chosen by each method etc. ) Does NOCTA re-sample the same feature a lot of times?
- minor: $\mathcal{V} \to \mathcal{V}_t$ since the set changes for each time-step


Overall, the paper tackles a strong and open problem, but current evaluation needs more in-depth analysis-- I am leaning weak reject, though I am willing to raise my score if my concerns are addressed.

**Questions:**

Mentioned along with weaknesses.

---

### Official Review · Reviewer_73S5 · 2025-11-11

**Soundness:** 2
**Presentation:** 3
**Contribution:** 2
**Rating:** 4
**Confidence:** 3

**Summary:**

This paper tackles longitudinal active feature acquisition (AFA), where an agent must sequentially decide which features to acquire at which timepoints to balance prediction accuracy against acquisition costs. The authors formulate a non-greedy objective that accounts for joint informativeness of future acquisitions and propose two methods to optimize it: (1) NOCTA-NP, a non-parametric approach using nearest neighbors in learned embedding space, and (2) NOCTA-P, a parametric value network approach. The authors prove NOCTA lower-bounds the optimal MDP policy and demonstrate superior performance on synthetic and medical datasets (ADNI, OAI) compared to RL-based and greedy baselines.

**Strengths:**

Following are the strengths of the paper:

1. This paper addresses an important and well motivated problem of Longitudinal Active Feature Acquisition (AFA) under temporal constraints especially in healthcare and other resource-constrained applications.

2. Also, the formulation of the objective (Eq. 1) elegantly captures the joint utility of future acquisitions rather than myopic per-feature gains. In addition to this the iterative reassessment (Algorithm 1) after each acquisition is practical and mirrors real clinical workflows.

3. Paper also demonstrates strong empirical performance of their methodology under different datasets when compared to other baselines.

**Weaknesses:**

Following are the main weakenesses of the paper:

1. The procedure seems computationally intensive.
    - The action space is exponential $2^{ML}$ and optimizing the subset selection each time seems computationally hard.
    - Empirically, the paper uses uniform random sampling of only 1000 candidates to address this. No justification for why this heuristic should find near-optimal solutions in all the settings.

2.  Empirical analysis needs to be expanded to check the scalability and robustness of the framework. Currently only tested on M=2 to M=27 features, $L \leq 12$ timepoints.

4. Embedding design lacks clarity (Eq. 4-7) and theoretical justification.
   - Why does each candidate $v_l$ induces uniform distribution over each selected feature-time pair?
   - Why should Jensen-Shannon divergence on top-κ weighted candidates be the right metric?

5. Theoretical Analysis seems limited. Theorem 1 only says ``NOCTA lower bounds the value of the optimal longitudinal AFA MDP policy'' but doesn't characterize tightness.
  - Further, how well NOCTA is being achieved using the given procedures is not characterized.
  - There is no empirical analysis of gap between NOCTA and true optimal policy.

**Questions:**

1. What happens if more principled discrete optimization for subset selection are used in the algorithms (e.g. beam search, Gumbel-softmax relaxations, or other methods)?

2. Why $\kappa=5$ in NOCTA-NP implementation? How sensitive is performance to this choice?

3. Hyperparameter $\alpha$ varies widely across datasets/tasks (e.g., 5e-5 to 0.03)—how to select in practice?

4. Can you empirically estimate the gap between NOCTA and optimal (perhaps on small instances where exhaustive search is feasible)?

---

### Note · Authors · 2025-11-14

I have read and agree with the venue's withdrawal policy on behalf of myself and my co-authors.